# Impact evaluation of a digital health platform empowering Kenyan women across the pregnancy-postpartum care continuum: A cluster randomized controlled trial

Rajet Vatsa[1]*, Wei Chang[2], Sharon Akinyi[3], Sarah Little[3], Catherine Gakii[4], John Mungai[4], Cynthia Kahumbura[3], Anneka Wickramanayake[3], Sathyanath Rajasekharan[3], Jessica Cohen[5‡], Margaret McConnell[5‡]

1 Harvard/MIT MD-PhD Program, Harvard Medical School, Boston, Massachusetts, United States of America, 2 Africa Chief Economist Office, World Bank, Washington, DC, United States of America, 3 Jacaranda Health, Nairobi, Nairobi County, Kenya, 4 Innovations for Poverty Action, Nairobi, Nairobi County, Kenya, 5 Department of Global Health and Population, Harvard T. H. Chan School of Public Health, Boston, Massachusetts, United States of America

‡ These authors are joint senior authors on this work.
* rajet_vatsa@hms.harvard.edu

**Data Availability Statement:** Data cannot be shared publicly because of potentially sensitive patient information, and the ethical approval under

## Abstract

### Background

Accelerating improvements in maternal and newborn health (MNH) care is a major public health priority in Kenya. While use of formal health care has increased, many pregnant and postpartum women do not receive the recommended number of maternal care visits. Even when they do, visits are often short with many providers not offering important elements of evaluation and counseling, leaving gaps in women's knowledge and preparedness. Digital health tools have been proposed as a complement to care that is provided by maternity care facilities, but there is limited evidence of the impact of digital health tools at scale on women's knowledge, preparedness, and the content of care they receive. We evaluated a digital health platform (PROMPTS (Promoting Mothers in Pregnancy and Postpartum Through SMS)) composed of informational messages, appointment reminders, and a two-way clinical helpdesk, which had enrolled over 750,000 women across Kenya at the time of our study, on 6 domains across the pregnancy-postpartum care continuum.

### Methods and findings

We conducted an unmasked, 1:1 parallel arm cluster randomized controlled trial in 40 health facilities (clusters) across 8 counties in Kenya. A total of 6,139 pregnant individuals were consented at baseline and followed through pregnancy and postpartum. Individuals recruited from treatment facilities were invited to enroll in the PROMPTS platform, with roughly 85% (1,453/1,700) reporting take-up. Our outcomes were derived from phone surveys conducted with participants at 36 to 42 weeks of gestation and 7 to 8 weeks post-childbirth. Among eligible participants, 3,399/3,678 women completed antenatal follow-up and

which the data were collected indicated the data were to be accessible to only the study team and would not be transmitted elsewhere. For further information, please contact the Amref Health Africa Ethics and Scientific Review Committee (esrc.kenya@amref.org) or the Harvard Chan School Office of Regulatory Affairs and Research Compliance (orarc@hsph.harvard.edu). The code used to conduct our evaluation is available on GitHub (https://github.com/RVatsa96/prompts-evaluation).

**Funding:** This study was made possible by the generous support of the American people through the United States Agency for International Development (USAID)'s Development Innovation Ventures (DIV) program (https://www.usaid.gov/DIV). SR and the team at Jacaranda Health were granted DIV funding under fixed award FAIN 7200AA20FA00025. The contents of this study are the sole responsibility of the authors and do not necessarily reflect the views of USAID or the United States Government. SR and the team at Jacaranda Health were also granted funding from the Bill & Melinda Gates Foundation (BMGF; https://www.gatesfoundation.org/) under awards INV-018440 and INV-043251. RV was supported by funding from the National Institute of General Medical Sciences (NIGMS) for his MD-PhD training (https://www.nigms.nih.gov/) under award numbers T32GM007753 and T32GM144273. The contents of this study are solely the responsibility of the authors and do not necessarily represent the official views of the National Institute of General Medical Sciences or the National Institutes of Health (NIH). USAID, BMGF, NIGMS, and NIH had no role in study design, data collection and analysis, decision to publish, or preparation of the study manuscript. Jacaranda Health (https://jacarandahealth.org/) contributed some independent funds to support the study as well. CK and SR are Co-Executive Directors; AW is Director of Research, Evaluation and Design; SL is Global Medical Director; and SA is a Research Manager at Jacaranda Health. Jacaranda Health had no other role in study design, data collection and analysis, decision to publish, or preparation of the study manuscript.

**Competing interests:** AW, CK, SA, SL, and SR are employees of Jacaranda Health.

**Abbreviations:** AI, artificial intelligence; ANC, antenatal care; BPCR, birth preparedness and complication readiness; FWER, family-wise error rate; ITT, intention-to-treat; MNH, maternal and newborn health; PNC, postnatal care; PROMPTS, Promoting Mothers in Pregnancy and Postpartum Through SMS; RCT, randomized controlled trial;

5,509/6,128 women completed postpartum follow-up, with response rates of 92% and 90%, respectively. Outcomes were organized into 6 domains: knowledge, birth preparedness, routine care seeking, danger sign care seeking, newborn care, and postpartum care content. We generated standardized summary indices to account for multiple hypothesis testing but also analyzed individual index components.

Intention-to-treat analyses were conducted for all outcomes at the individual level, with standard errors clustered by facility. Participants recruited from treatment facilities had a 0.08 standard deviation (SD) (95% CI [0.03, 0.12]; $p = 0.002$) higher knowledge index, a 0.08 SD (95% CI [0.02, 0.13]; $p = 0.018$) higher birth preparedness index, a 0.07 SD (95% CI [0.03, 0.11]; $p = 0.003$) higher routine care seeking index, a 0.09 SD (95% CI [0.07, 0.12]; $p < 0.001$) higher newborn care index, and a 0.06 SD (95% CI [0.01, 0.12]; $p = 0.043$) higher postpartum care content index than those recruited from control facilities. No significant effect on the danger sign care seeking index was found (95% CI [−0.01, 0.08]; $p = 0.096$).

A limitation of our study was that outcomes were self-reported, and the study was not powered to detect effects on health outcomes.

## Conclusions

Digital health tools indicate promise in addressing shortcomings in pregnant and postpartum women's health care, amidst systems that do not reliably deliver a minimally adequate standard of care. Through providing women with critical information and empowering them to seek recommended care, such tools can improve individuals' preparation for safe childbirth and receipt of more comprehensive postpartum care. Future work is needed to ascertain the impact of at-scale digital platforms like PROMPTS on health outcomes.

## Trial Registration

ClinicalTrials.gov ID: NCT05110521; AEA RCT Registry ID: R-0008449

---

## Author summary

### Why was this study done?

- In Kenya, quality of maternal and newborn health (MNH) care is key to mitigating persistently high rates of morbidity and mortality, yet evidence suggests that quality is lacking.

- Digital health tools may complement formal facility-based care, with the potential to improve antenatal and postpartum knowledge, preparedness, and use of recommended health care.

- Limited rigorous evidence exists in low- or middle-income settings on the impact of at-scale digital health tools on pregnant and postpartum women's knowledge, preparedness, health behaviors, and the content of care they receive.

SD, standard deviation; SMS, Short Message Service.

## What did the researchers do and find?

- We evaluated a digital health platform supporting pregnant and postpartum women called PROMPTS (Promoting Mothers in Pregnancy and Postpartum Through SMS), which consists of informational messages, appointment reminders, and a clinical helpdesk.

- We conducted a cluster randomized controlled trial across 40 health facilities in 8 Kenyan counties, in which 6,139 pregnant women were enrolled, with those recruited from treatment facilities additionally invited to enroll in PROMPTS.

- Overall, we found that this encouragement to enroll in PROMPTS led to a range of modest improvements across the pregnancy-postpartum continuum, including in knowledge, birth preparedness, care seeking, newborn care behaviors, and the content of postpartum care.

## What do these findings mean?

- Digital health tools can complement facility-based antenatal and postpartum care by providing important information about recommended care, danger signs, birth preparedness, and other factors that can improve maternal and newborn outcomes.

- These tools may be particularly effective in improving postpartum care for both mothers and newborns, an area in Kenya that has historically lagged behind improvements in antenatal care attendance and facility-based childbirth.

- Our study leveraged self-reported outcomes and was not designed to detect effects on health outcomes; our findings should be interpreted with these limitations in mind.

## Introduction

While maternal and newborn health (MNH) outcomes have improved globally, progress over the past decade has been slow or stagnant in many low- and middle-income settings. In Kenya, for example, the maternal mortality rate remains well over 5 times the United Nations' Sustainable Development Goal [1]. Stillbirth and neonatal mortality rates in Kenya have also been stagnant for much of the 21st century, though declines in neonatal mortality have been reported in recent years [2,3]. In addition, high rates of maternal and neonatal severe morbidity persist across sub-Saharan Africa, including in Kenya [4–7].

Accelerating improvements in MNH care to reduce mortality and morbidity is a major public health priority in Kenya. In recent decades, there have been substantial shifts in utilization of formal maternity care: 90% of pregnant women in Kenya receive some form of antenatal care (ANC) from a medical professional, and over 80% of births occur at health facilities, nearly double the rate from 2003 [8–10]. Nevertheless, 40% of pregnant women do not receive the nationally recommended minimum of 4 ANC visits, and just over 50% receive any postpartum care, rates that have remained stable over the past 2 decades [10–12]. Even when visits happen, studies from low- and middle-income settings have revealed that time spent with providers is often short, frequently lasting less than 5 min [13]. A recent multicountry survey with

respondents from Kenya relatedly found that approximately 1/3 of patients reported inadequate time with their providers [14].

Amidst this backdrop of mixed advances towards adequate formal care utilization, research increasingly highlights the need for improvements in the quality of care [14–16]. However, existing evidence from Kenya suggests that the majority of women do not receive a minimally adequate standard of antenatal and postpartum care [17]. For example, many individuals are not offered a host of essential care components, with systematic assessments revealing that—across poverty levels—fewer than 60% of women were comprehensively examined or counseled on critical topics like pregnancy complications and birth preparation [14,17]. Thus, many pregnant and postpartum women remain ill-informed about key aspects of their health and are not empowered to seek recommended care—for both routine and acute cases—at the right place and time [10,18–21]. When individuals are underequipped with such knowledge, it can result in delays in the decision to seek care. These delayed care decisions are often referred to as the "first delay" in the Three Delays Model of MNH care and are associated with increased maternal and neonatal morbidity and mortality [18,22–24].

Historically, such challenges have been addressed both at the health system level and at the level of individual pregnant and postpartum women. Health system-level strategies predominantly entail some form of provider training; however, these approaches can be costly and have demonstrated mixed success, with modest effect sizes and issues with sustainability at scale [14,25–28]. Individual-level strategies, on the other hand, include programs that mitigate barriers to care seeking (e.g., support groups, transport vouchers, and insurance) and informational tools that promote health literacy and receipt of recommended care [27,28]. Such approaches have demonstrated success in improving care utilization but can be challenging to integrate with the formal MNH care system. Moreover, to date, they have underemphasized outcomes for mothers in the postpartum period [27,29,30].

Digital health tools, however, offer unique promise as an individual-level strategy to complement care provided in the formal sector and to close the gaps described previously across the pregnancy-postpartum continuum. With high rates of mobile phone penetration across low-income countries, the prevalence of such interventions has increased in recent decades [31]. To date, digital health tools targeting MNH care have primarily sought to promote recommended health behaviors and care seeking through Short Message Service (SMS) or voice-based education and nudges [30,32,33]. Other tools aim to connect women with care providers via digital communication platforms (e.g., digital helpdesks) [32]. In multiple systematic reviews, existing tools have been associated with improvements in utilization of ANC, facility-based birth, skilled birth attendance, and newborn vaccination, though few studies have examined or identified an impact on knowledge, preparedness, or content of care, particularly for mothers in the postpartum setting [30–32].

With the recent rise of artificial intelligence (AI) as well, such tools can be layered with AI to efficiently triage problems and deliver targeted education to pregnant and postpartum women [34]. Thus, if integrated effectively, digital health tools have the potential to reduce strain on the formal health care system, enhance access to critical information, and empower individuals to receive the right care at the right place and time. Ultimately, well-powered, randomized evidence from low- and middle-income settings on the impact of digital health tools on a broad set of MNH outcomes is critical to understand how these programs could perform in implementation settings [31,32,35].

In this paper, we describe a parallel arm cluster randomized controlled trial (RCT) carried out in 40 health facilities across Kenya to test the impact of a digital health platform called PROMPTS (Promoting Mothers in Pregnancy and Postpartum Through SMS). PROMPTS was developed by Jacaranda Health, a leading MNH nonprofit organization in Kenya, to

"empower women to seek care at the right time and place and give them greater agency in the health system" [36]. Through collaborations with regional and national government officials, over 900 health facilities across Kenya had begun enrolling pregnant and postpartum women onto the platform, with over 750,000 enrollees by November 2021, the start of our study period. While a prior randomized trial evaluated a precursor to the PROMPTS platform, the study took place across 3 facilities in a single county, enrolled fewer than 1,000 women and was focused only on the postpartum setting [37]. Accordingly, our study contributes to the literature by rigorously evaluating the impact of an individual-level, digital health solution for MNH care that has been implemented at scale in a lower-middle income setting. In addition, we expand beyond the narrow focus on antenatal and postpartum care seeking and examine the effect of PROMPTS on knowledge, preparedness, and content of care, for both pregnant and postpartum women and their newborns.

## Methods

### Study design, intervention, and participants

We conducted an unmasked, 1:1 parallel arm cluster RCT in 40 health facilities across 8 counties in Kenya. Treatment facilities received 2 different interventions, one targeted towards pregnant and postpartum women and the other targeted towards maternity care providers. The first intervention was that women receiving ANC at treatment facilities were invited to enroll in PROMPTS. The second intervention was a nurse mentorship program designed to increase and sustain providers' knowledge and skills in basic and emergency obstetric and newborn care. We evaluated these components separately, focusing here on the impact of facilities' offering PROMPTS on pregnant and postpartum women's knowledge, preparedness, care utilization, health behaviors, and content of care, using longitudinal surveys of participants across the antenatal and postpartum settings. The nurse mentorship program was focused largely on quality of emergency care during childbirth and was not intended to influence the outcomes analyzed here. Further, 97% of the PROMPTS sample delivered in study facilities before rollout of the nurse training was complete. The nurse training intervention is being evaluated in separate work.

Pregnant women receiving ANC at treatment facilities were invited to enroll in PROMPTS during in-person interactions with onsite enumerators. In control facilities, PROMPTS was not systematically introduced to participants, either during interactions with study enumerators or through standard processes of care. However, women enrolled from control facilities were unrestricted from independently enrolling in PROMPTS and may have done so through other channels, such as learning about the platform from personal contacts or providers exposed to it. Evaluation outcomes were measured through longitudinal surveys with women during the antenatal and postpartum period (**Fig 1**).

We preregistered our evaluation of the complete intervention package on the United States Clinical Trials Registry (NCT05110521) and the American Economic Association's RCT Registry (AEARCTR-0008449). Our study is reported according to the Consolidated Standards of Reporting Trials (CONSORT) for cluster-randomized trials [38,39]; see **S1 Checklist** and **S2 Checklist**.

**Intervention.** PROMPTS consists of informational messages, appointment reminders, and a two-way clinical helpdesk. SMS messages in English or Swahili—determined based on language preferences solicited at enrollment—are sent to enrollees based on their gestational age and are designed to influence relevant health behaviors. The content that participants receive has been rigorously tested by Jacaranda Health with intended beneficiaries and accordingly adapted to ensure broad accessibility, comprehension, and convenience (e.g., suitable

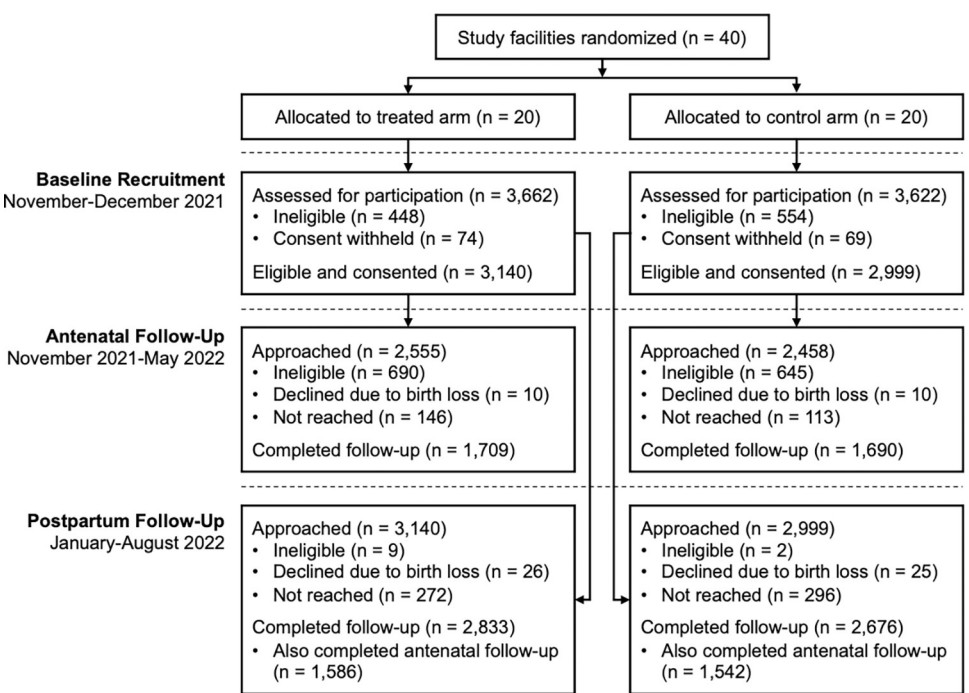

**Fig 1. Retention at antenatal and postpartum follow-up of the cohort of pregnant women recruited at baseline.**
This figure synthesizes the baseline recruitment of pregnant women who visited health facilities for antenatal care and tracks their retention at antenatal and postpartum follow-up, which occurred in the final weeks of pregnancy (i.e., around 36–42 weeks of gestation) and 7 to 8 weeks post-childbirth, respectively. Women recruited at or beyond 36 weeks of gestation at baseline were not approached for antenatal follow-up, while participants whose pregnancy had ended with birth of a living newborn were approached but ineligible. Meanwhile, for postpartum follow-up, all eligible and consented participants at baseline were approached and only a handful were ineligible due to being fewer than 7 weeks postpartum.

frequency and timing of messages). Depending on the timing of enrollment, participants are sent roughly 10 to 40 messages per month during pregnancy and roughly 10 messages per month during the first year postpartum. The PROMPTS platform also includes an AI-enabled helpdesk that assesses, triages, and responds to questions. The helpdesk first runs messages through a natural language processing pipeline to assign each message an intent and priority level. Accordingly, the helpdesk sends automated guidance (roughly 86% of cases) or escalates messages to trained clinical agents who staff the helpdesk 24/7 and respond within 1 min for high-priority cases [36,40]. Finally, PROMPTS users are queried about their maternal care content and experience and the aggregated data is shared with facilities and health system managers to build accountability and target system improvements. PROMPTS is free for participants, runs via SMS on both basic and smartphones, and is primarily rolled out through public health facilities. At the time of our study, PROMPTS was solely a text-based platform and did not offer audio support for low-literacy or sight-impaired women.

**Health facility eligibility.**   Health facilities were eligible for study inclusion if they were owned by the government or a faith-based organization and had an average of 50 to 400 normal (i.e., unassisted) vaginal births per month, based on records from the Kenya Health Management Information System. Facilities were also eligible based on having no known ongoing mobile health or quality-improvement research initiatives.

**Participant recruitment and eligibility.**   Pregnant women were recruited by onsite enumerators at health facilities during ANC and were followed up by phone during pregnancy and postpartum. At baseline, all women who arrived at the facility for ANC were assessed for

participation, eligibility, and consent, subject to enumerator availability. To be eligible, participants had to have access to a mobile phone (basic phone or smartphone), be at least 15 years old, and be at least 16 weeks pregnant.

### Randomization and masking

Randomization was conducted at the facility level given that enrollment into PROMPTS is designed to take place at health facilities (**Fig 2**). Candidates for randomization included eligible facilities in counties in which Jacaranda Health had a plan to expand. Facility locations were reviewed with the team at Jacaranda Health, and in select cases where 2 facilities were near one another (e.g., in the same sub-county), the facility with the lower level or volume was excluded. Facility-level randomization was stratified by tertile of monthly volume of normal vaginal births (see **S1 Text**). Given the nature of the intervention, facilities were unblinded and informed of their random assignment prior to the start of study activities.

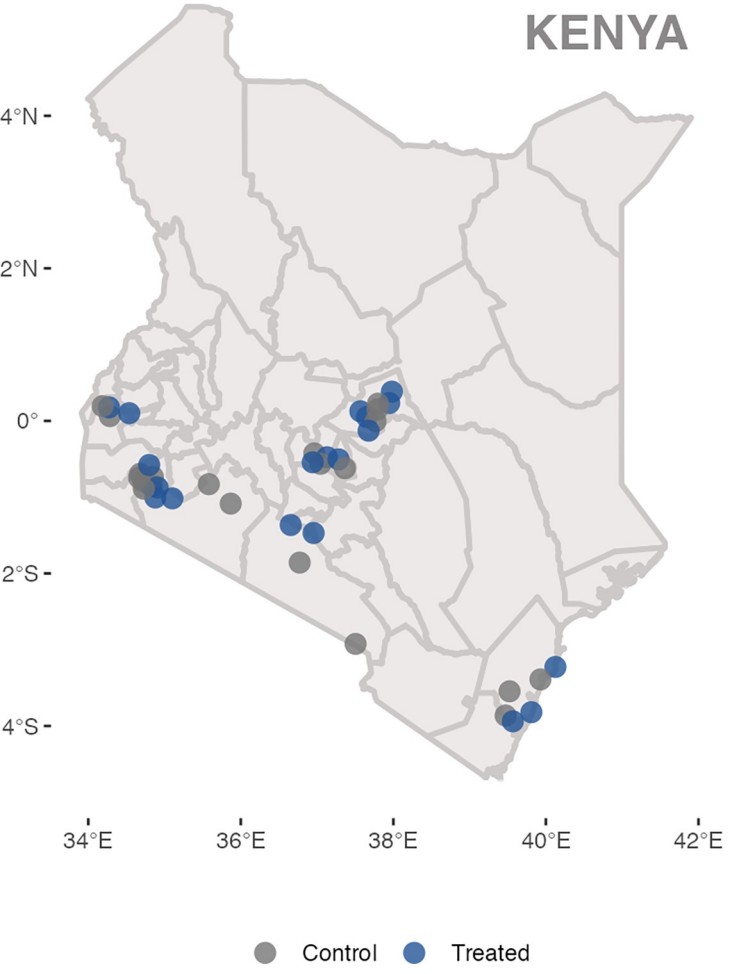

**Fig 2. Map of study facilities.** This figure displays the location of the 40 health facilities in the study. The study facilities were distributed across 8 counties (Kajiado [4 facilities], Kilifi [6], Kirinyaga [2], Kisii [8], Meru [8], Narok [4], Nyeri [4], and Siaya [4]). Facilities are color-coded by treatment (blue) vs. control (gray) status. For 33 of the 40 study facilities, the closest facility was 10+ kilometers away. For the remaining 7 facilities, the nearest facility belonged to the same study arm. The distance between each facility and the nearest facility of the opposite arm was, on average, 32 kilometers. Map data courtesy of https://open.africa/dataset/kenya-counties-shapefile/resource/0b78f25e-494e-4258-96b8-a3ab2b35b121; available under a Creative Commons Attribution license.

Ultimately, 37 of the 40 included study facilities were owned by the Ministry of Health of Kenya, and 3 were owned by faith-based organizations. The facilities were primarily Level 4 hospitals (32 facilities), though 5 were Level 3 health centers and 3 were Level 5 county referral hospitals. For context, Kenya's health care system is structured in 6 levels, with higher-level facilities designed to absorb referrals for complicated cases encountered by lower levels. Levels 1 to 5 facilities are managed at the (sub)-county level, whereas Level 6 facilities are managed at the national level.

## Procedures

**Data collection.** All participants completed an in-person informed consent process followed by a baseline survey. A subsample of women (those who were enrolled prior to their 36th week of gestation) were contacted by mobile phone for a follow-up antenatal survey in the final weeks of pregnancy (i.e., around 36 to 42 weeks of gestation). All women surveyed at baseline were contacted by mobile phone for a follow-up postpartum survey, 7 to 8 weeks after childbirth (**Fig 1**).

Baseline and follow-up surveys asked questions about demographic and health information, health care seeking, knowledge of maternal and newborn danger signs, health practices and preparedness, and content of care. In cases of stillbirth, miscarriage, or infant loss at antenatal or postpartum follow-up, participants were offered resources to connect with trained psychological counselors and to abstain from being interviewed. In cases where participants elected to continue being interviewed, a shortened survey was administered. Baseline data collection lasted from November 8, 2021 to December 10, 2021; antenatal follow-up data collection lasted from November 24, 2021 to May 25, 2022; and postpartum follow-up data collection lasted from January 5, 2022 to August 24, 2022. Participants received an SMS credit of 100 Kenyan shillings (slightly <1 US dollar in 2022) per survey completed.

**Enrollment in PROMPTS.** Immediately following the baseline survey, women in treatment facilities were invited to enroll in PROMPTS by study enumerators. At the time of consent and enrollment, women were provided with high-level information about the platform, an overview of the program's goals to mitigate delays in care seeking, and an assurance of the study team's commitment to data protection and privacy. Those who agreed to participate started receiving messages as soon as their contact information was shared with Jacaranda Health, typically within 24 h of enrollment.

## Outcomes

We organized individual-level outcomes into 6 domains: knowledge, birth preparedness, routine care seeking, danger sign care seeking, newborn care, and postpartum care content. **Fig 3** offers a conceptual framework motivating the assessment of outcomes in these domains. For each domain, we generated summary indices to account for multiple hypothesis testing, as detailed in Anderson (2008) [41]. Specifically, within each index, we ensured directional consistency of normalized component outcomes and calculated their inverse-covariance-weighted average. Thus, highly correlated components received lower weight, emphasizing index components that offered unique information. Though most component measures were preregistered outcomes, in a few instances, we included exploratory outcomes that were not preregistered but were strongly connected to the program's theory of change.

Preregistered versus exploratory component measures are clearly denoted throughout, as well as summarized in **S1 Table** and **S2 Text**. **S1 Table** and **S2 Text** also delineate whether preregistered measures were primary or secondary, highlighting the range of domains that PROMPTS attempts to influence, as opposed to a singular primary outcome. Of note, a

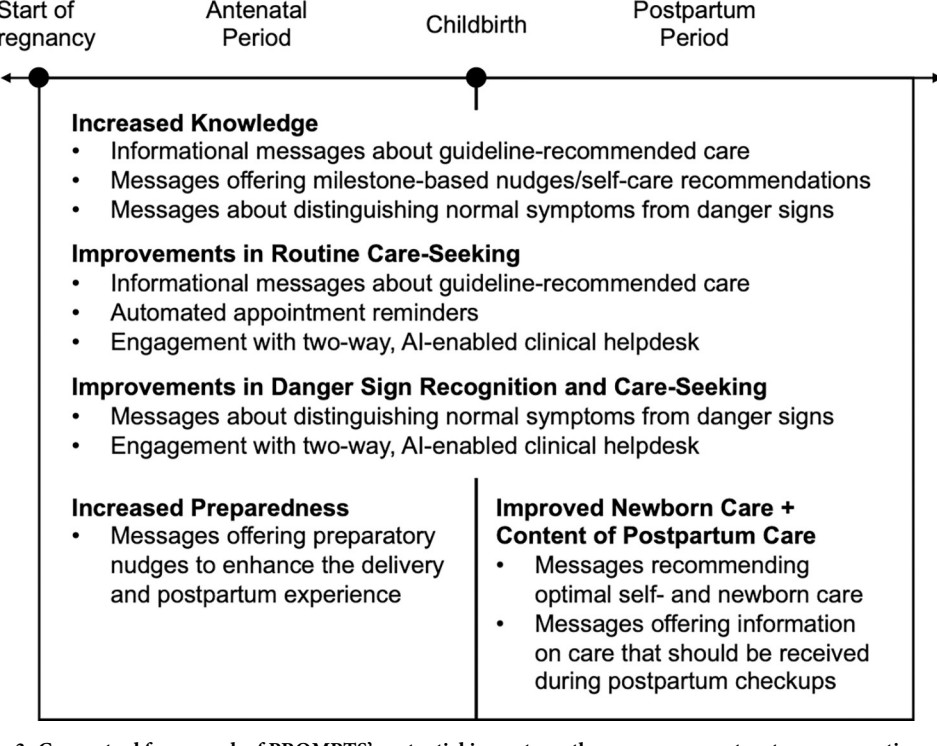

**Fig 3. Conceptual framework of PROMPTS's potential impacts on the pregnancy-postpartum care continuum.**
This figure offers a conceptual framework of the potential impacts PROMPTS could have across the pregnancy-postpartum care continuum. During both the antenatal and postpartum periods, improvements in knowledge (e.g., through informational messages) could lead to improvements in routine care seeking and danger sign recognition/care seeking. During the antenatal period, messages containing preparatory nudges may enhance preparation for childbirth, while during the postpartum period, messages containing care recommendations and content of care reminders may enhance newborn care and postpartum care content for infants and mothers. PROMPTS is also comprised of automated appointment reminders and a two-way clinical helpdesk, which may influence routine and danger sign care seeking.

handful of preregistered measures are not reported on in the main manuscript. **S2 Text** describes these measures along with a justification for their exclusion. Finally, **Table 1** presents a detailed set of PROMPTS messages that correspond to several of the outcomes that we evaluated.

Within the knowledge domain, we constructed an index composed of 4 preregistered outcomes: the shares of antenatal, postpartum, and neonatal danger sign knowledge questions answered correctly and the number of signs of labor listed, without prompting (see **S3 Text**).

In terms of birth preparedness, we constructed an index composed of 2 preregistered outcomes and one exploratory outcome. The first preregistered outcome was the total number of items reported to have been completed, without prompting, in preparation for childbirth (see **S3 Text**). The second was whether a participant had a plan to breastfeed within 1 h of childbirth. Given PROMPTS's objective to mitigate delays in care seeking, we also explored timeliness of facility arrival for childbirth, specifically whether a participant arrived late (i.e., within 2 h of childbirth). We posited that this outcome may shift as a reflection of changes in preparedness.

Within the domain of routine care seeking, we constructed an index composed of 6 preregistered outcomes and 2 exploratory outcomes. The preregistered outcomes included the total number of antenatal care visits attended; a binary indicator denoting receipt of postnatal care (PNC), within 6 weeks of childbirth, where one's own health was discussed; a binary indicator

**Table 1. Outcome-related summary of SMS content delivered to PROMPTS enrollees.**

| Outcome | Pertinent PROMPTS Message |
| --- | --- |
| **Knowledge** | |
| Labor signs | "Do you know the signs of labor? This will feel like lower abdominal pain that is rhythmic and increasing in intensity, low back pain or a change in discharge to brownish or a mucus-like discharge with blood. If you experience these or heavy bleeding, or breaking of water, go to the hospital immediately…" |
| Knowledge of normal symptoms vs. danger signs | "Nausea and vomiting are common in pregnancy… It is important to distinguish if your vomiting is due to normal morning sickness… or something more serious… Reasons you should contact your health care provider are if the vomiting is persistent and leading to weight loss, if it is bloody or associated with fever or diarrhea." |
| **Birth Preparedness** | |
| Items done in preparation for childbirth | "Delivering in a hospital is the safest option for you and your baby. Have a packed bag for you and your baby before your due date. Ensure you have someone who can give you support in case you go into labour unexpectedly. Save some money to cater for transport in case you go into labor unexpectedly. Ensure you already have a choice of hospital where you will deliver." |
| Plan to breastfeed within 1 hour of childbirth | "Breastfeeding your baby is one of the best things you can do for your newborn! You should start breastfeeding within 1 hour after you deliver…" |
| Late facility arrival for childbirth $^\vee$ | No messages directly pertaining to timing of facility arrival |
| **Routine Care Seeking** | |
| Number of ANC visits attended / guideline-recommended ANC receipt $^\vee$ | "Did you know that you should visit the clinic at least 4 times during your pregnancy? The visits will help ensure your baby is growing well and that you will have a safe delivery." |
| PNC visit within 6 weeks of childbirth during which mother's own health discussed | "It is very important that you return to the hospital 2 weeks after delivery or on your appointment date for postnatal care… You and your baby will be examined to ensure you are healthy after delivery…" |
| Guideline-recommended PNC receipt $^\vee$ | No messages specifically called out the recommended number of PNC visits, though messages emphasized the importance of attending scheduled PNC appointments. |
| Facility-based childbirth | "Delivering in a hospital is the safest option for you and your baby… Ensure you already have a choice(s) of hospital where you will deliver." |
| **Danger Sign Care Seeking** | |
| Care seeking in response to antenatal danger signs | "Watch for danger signs in pregnancy. Are you experiencing vaginal bleeding, severe nausea and vomiting, fever (>37.8˚C), other flu-like symptoms or severe abdominal pain? If you are seeing any of these danger signs… go to the nearest hospital immediately." |
| Care seeking in response to postpartum danger signs | "Are you feeling dizzy, faint, having difficulty breathing, a headache that won't go away, severe upper abdominal pain, or difficulty seeing? If you are seeing any of these danger signs… go to the nearest hospital immediately." |
| Care seeking in response to neonatal danger signs | "If your baby is not breastfeeding every 2–3 hours, is having convulsions, is not passing urine every 8 hours, is not able to be woken up to feed, is breathing with the chest going in instead of out, or has a fever, she/he is showing a danger sign. Are you seeing any of these danger signs? If yes… go to the nearest hospital immediately." |
| **Newborn Care** | |
| Exclusive breastfeeding | "Breast milk is so good for your baby! You should breastfeed without introducing other foods for 6 complete months of your baby's life…" |

(*Continued*)

**Table 1.** (Continued)

| Outcome | Pertinent PROMPTS Message |
|---|---|
| Safe sleep positioning | "Put your baby to sleep on her/his back. This is the safest way for your baby to sleep through the night." |
| Frequent engagement via singing/talking | "No matter how many other children you have and whether you have a girl or a boy, talk and sing to your baby often, and give plenty of hugs. All babies need equal attention." |
| **Postpartum Care Content** | |
| Physical exam for mother conducted [∀] | "Hi Mum, please remember that you have a postnatal checkup today. During your visit, the provider should examine you and your baby, ask about your breastfeeding progress, and discuss family planning options with you." |
| Physical exam for newborn conducted [∀] | |
| Family planning discussed [∀] | |
| Immunizations for newborn administered [∀] | "Hi mum. . . At the 6 weeks visit, your baby should have received 2 oral vaccines: Polio and Rota Virus. . . Your baby should also receive 2 injections: DTP/Pentavalent and Pneumonia vaccine. . ." |
| Cervical cancer screening offered [∀] | No messages pertaining to cervical cancer screening but inquired about in the same section of the postpartum follow-up survey as the preceding outcomes. |

[∀] An unregistered exploratory outcome.

ANC, antenatal care; DTP, diphtheria-tetanus-pertussis; PNC, postnatal care.

denoting that childbirth occurred in a hospital or other formal health center; and 3 separate binary indicators denoting whether a participant sought antenatal (within past month), postpartum, or neonatal medical advice or treatment from a health care professional. In exploratory analyses of guideline-based care, we generated binary indicators denoting receipt of the nationally recommended number of ANC and PNC visits, with Kenya's national guidelines recommending at least 4 ANC visits over the course of one's pregnancy and at least 2 PNC visits during the first 6 weeks postpartum, following the first postpartum week [42].

Within the domain of care seeking for danger signs, we constructed an index composed of 3 preregistered outcomes, corresponding to each of 3 categories of severe danger signs: antenatal, postpartum, and neonatal. For each category, we assessed whether participants reported seeking medical advice or treatment from a health care professional in response to at least a single danger sign (see **S3 Text**).

Within the domain of newborn care, we constructed an index composed of 3 preregistered outcomes, namely binary indicators denoting whether participants reported exclusively breastfeeding, always putting their newborn to sleep through the night on their back, and singing or talking to their newborn many times during the preceding day.

Finally, within the domain of postpartum care content, we constructed an index composed of 6 exploratory outcomes: whether participants, during at least 1 PNC visit, reported that their health was discussed with a provider, that their provider conducted a physical exam for them, that their provider discussed family planning with them, that their provider offered them cervical cancer screening, that their provider conducted a physical exam for their newborn, and that their provider offered immunizations to their newborn.

## Statistical analysis

Statistical analyses were conducted using Stata/MP 17.0 and R 4.2.3. The impacts of the intervention are presented as intention-to-treat (ITT) analyses, representing the impact of the offer of PROMPTS on outcomes. Impacts on individual-level binary and continuous outcome variables were estimated via ordinary least squares using linear (probability) models. With respect

to the summary indices, the magnitudes of the impacts estimated can be interpreted as standard deviation changes in the respective domain in the treatment group compared to the control group. In our unadjusted regression specification, outcomes were regressed on a treatment indicator denoting whether an individual was recruited from a treatment facility and a stratification variable denoting the recruitment facility's baseline normal vaginal birth volume tertile. To improve the precision of our average treatment effect estimates, we also estimated adjusted models in which we added a set of baseline individual- and facility-level covariates that we hypothesized would best account for residual variation in our outcomes of interest. These included maternal age, gestational age, an indicator for first pregnancy, an indicator for secondary school attainment, an indicator for adequate ANC (accounting for gestational age), an indicator for previous receipt of SMS messages from one's county offering pregnancy advice, the recruitment facility level, and the recruitment facility county. Adjusted effect estimates are highlighted and discussed in the main text.

Given the facility-level randomization, standard errors were adjusted for clustering using clustered sandwich estimators, as detailed in Zeileis, Köll, and Graham (2020); all results are presented as point estimates with corresponding 95% confidence intervals and $p$-values [43]. To account for our multiple domains and control the family-wise error rate (FWER) at 0.05, we report Holm–Bonferroni-adjusted $p$-values for each summary index.

**Sensitivity analysis.** While the summary indices offer one strategy to aggregate outcomes and limit the number of hypotheses tested, we also present estimates for each domain's component measures. While we report unadjusted $p$-values for the index components in the main manuscript, we provide a side-by-side comparison of unadjusted and Holm–Bonferroni-adjusted $p$-values in **S2 Table**. The latter control the FWER for each domain at 0.05.

**Sample size and power.** The sample size of women who consented at baseline (see **Results**) exceeded the enrollment goal of 4,800 women (roughly 120 women per study facility), accounting for the assumption that ≥80% of women in treatment facilities would enroll in PROMPTS and that ≥60% of those women would be successfully engaged through endline: 120×0.8×0.6≈58 assessable women per study facility. This target was set to detect a minimum 7.0 percentage point (ppt) increase (11% relative increase) in the rate of PNC receipt within 6 weeks of childbirth, assuming a reference rate of 61% (based on internal data from Jacaranda Health), intracluster correlation of 0.01, one-sided type 1 error rate of 5%, and power of 80%. It was also set to detect a minimum 0.25 visit increase (6% relative increase) in the total number of ANC visits, assuming a reference mean of 4.12 and standard deviation (SD) of 1.73 (based on the 2014 Kenya Demographic and Health Survey) [44]. We focused our power calculations on these 2 preregistered outcomes, for which we had administrative data.

## Ethics and safety statement

This study was licensed by the National Commission for Science, Technology, and Innovation in Kenya (NACOSTI/P/21/13369) and approved by the institutional review board and ethics and scientific review committee of the Harvard T.H. Chan School of Public Health (IRB21-1013; see **S1 Protocol**) and Amref Health Africa (P1047/2021). Written, electronic, or thumbprint-based informed consent was obtained from all participants. An emphasis on voluntary participation, participant confidentiality, and freedom to withdraw consent at any time was upheld throughout.

## Results

### Sample attrition and characteristics

At baseline, 7,284 women were approached across the 40 study facilities, 6,139 of whom (3,140 in the treatment arm; 2,999 in the control arm) were eligible and consented to participate in

the study (**Fig 1**). Only participants who were enrolled into the study prior to their 36th week of gestation (*N* = 5,013) were contacted for the antenatal follow-up survey. Of those 5,013 women, 1,335 were ineligible due to their current pregnancy having already ended (with a live birth) when contacted. In these cases, these women were recontacted 7 to 8 weeks post-childbirth for the postpartum follow-up survey, unless their pregnancy had ended over 6 weeks ago, in which case they were immediately eligible for postpartum follow-up; 3,399/3,678 women (1,709 in the treatment arm; 1,690 in the control arm) ultimately completed the antenatal follow-up survey, representing an 8% loss to follow-up among those eligible (**Fig 1**). At postpartum follow-up, all 6,139 women who consented at baseline were contacted, and only 11 were ineligible due to being less than 7 weeks postpartum; 5,509/6,128 women (2,833 in the treatment arm; 2,676 in the control arm) completed the survey, representing a 10% loss to follow-up among the eligible sample (**Fig 1**). At both anteparthum and postpartum follow-up, the attrition rate did not significantly differ between study arms (see **S3 Table**).

Baseline characteristics of the study sample overall and by study arm are presented in **Table 2**. The average age of participants was 26 years, with 82% (5,040/6,139) of the sample married or cohabitating and 65% (3,980/6,138) having completed secondary school or higher. Study participants were at varying stages of pregnancy, with an average gestational age of 30 weeks and around 80% (4,965/6,139) having received some form of prior ANC. Almost 90% (5,499/6,139) of the participants had their own mobile phone (as opposed to access to a close contact's phone), and 95% (5,846/6,139) could read English or Kiswahili without difficulty. Approximately 22% (1,327/6,096) reported having at least 1 high-risk condition (e.g., diabetes, pre-eclampsia, placenta previa) during the current pregnancy, as informed by a health provider, and among those with a prior pregnancy (66%; 4,040/6,139), 43% (1,722/4,040) reported history of either hypertension, pre-eclampsia, postpartum hemorrhage, preterm birth, stillbirth, neonatal death, or cesarean section. Finally, on average, participants correctly answered 68% of questions on a baseline knowledge assessment testing whether they would seek immediate medical care or watchfully wait in response to several potentially severe pregnancy symptoms. Sample characteristics were similar between study arms at baseline, as were the characteristics of those who completed the antenatal and postpartum follow-up surveys (**Tables 2**, **S4**, and **S5**).

## Intervention fidelity

Fidelity and uptake of the PROMPTS platform is summarized in **Table 3**, which reveals that among participants from treatment facilities, over 85% (1,453/1,700) reported receipt of messages offering pregnancy advice, while less than 10% (156/1,674) of participants from control facilities reported such receipt. Thus, recruitment from treatment facilities was associated with a 76 ppt (95% CI [73, 78]; *p* < 0.001) increase in the take-up of PROMPTS.

## Intervention impact

Women in the treatment arm had a 0.08 SD (95% CI [0.03, 0.12]; *p* = 0.002) higher knowledge index compared to those in the control arm, with impacts concentrated in the antenatal period (**Table 4**). Specifically, those in the treatment arm had a 3.6 ppt (95% CI [1.9, 5.4]; *p* < 0.001) higher antenatal danger sign knowledge score and could list 0.24 (95% CI [0.16, 0.32]; *p* < 0.001) more signs of labor than participants in the control arm. No statistically significant differences in the postpartum or neonatal danger sign knowledge indices were found.

Women in the treatment arm also had a 0.08 SD (95% CI [0.02, 0.13]; *p* = 0.018) higher birth preparedness index compared to those in the control arm (**Table 4**). This included completing 0.13 (95% CI [0.04, 0.21]; *p* = 0.003) more items in preparation for childbirth, such as

**Table 2. Baseline characteristics of eligible and consented cohort.**

| Baseline Characteristic | Overall (N = 6,139) | Control Arm (N = 2,999) | Treated Arm (N = 3,140) |
|---|---|---|---|
| Age (years) | 26.15 (5.89) | 25.95 (5.94) | 26.35 (5.84) |
| Completed secondary school education or higher ^ | 64.8% (3,980/6,138) | 63.6% (1,906/2,999) | 66.1% (2,074/3,139) |
| Ability to read Kiswahili or English without difficulty ^ | 95.2% (5,846/6,139) | 94.8% (2,843/2,999) | 95.6% (3,003/3,140) |
| Married or living together ^ | 82.1% (5,040/6,139) | 81.3% (2,437/2,999) | 82.9% (2,603/3,140) |
| Size of household | 3.81 (1.99) | 4.01 (2.13) | 3.62 (1.84) |
| Landowner ^ | 42.4% (2,596/6,128) | 47.1% (1,411/2,993) | 37.8% (1,185/3,135) |
| Access to an improved source of drinking water (e.g., piped water) ^ | 70.1% (4,303/6,139) | 69.3% (2,079/2,999) | 70.8% (2,224/3,140) |
| Access to an improved sanitation facility (e.g., flush toilet) ^ | 97.7% (6,000/6,139) | 97.2% (2,914/2,999) | 98.3% (3,086/3,140) |
| Access to a motor vehicle for travel to hospital ^ | 69.3% (4,255/6,139) | 69.6% (2,087/2,999) | 69.0% (2,168/3,140) |
| Time to travel from home to health facility (minutes) | 23.92 (18.54) | 24.64 (18.64) | 23.23 (18.43) |
| Worked for pay in last week ^ | 22.6% (1,390/6,139) | 22.3% (669/2,999) | 23.0% (721/3,140) |
| Easy access to KES 2,000 if treatment for illness needed in household ^ | 27.5% (1,674/6,098) | 27.3% (811/2,970) | 27.6% (862/3,128) |
| Access to own mobile phone ^ | 89.6% (5,499/6,139) | 88.0% (2,640/2,999) | 91.1% (2,859/3,140) |
| Frequent/daily use of mobile phone to send text messages ^ | 36.7% (2,255/6,139) | 33.5% (1,005/2,999) | 39.8% (1,250/3,140) |
| Previously received text message(s) offering pregnancy advice from county ^ | 5.0% (308/6,120) | 4.0% (120/2,990) | 6.0% (188/3,130) |
| Gestational age (weeks) | 29.69 (6.14) | 29.61 (6.16) | 29.76 (6.12) |
| Received prior ANC for current pregnancy ^ | 80.9% (4,965/6,139) | 80.7% (2,420/2,999) | 81.1% (2,545/3,140) |
| # ANC visits for current pregnancy | 1.93 (1.52) | 1.90 (1.48) | 1.97 (1.55) |
| Fraction of knowledge questions answered correctly | 0.68 (0.18) | 0.67 (0.18) | 0.69 (0.18) |
| Current pregnancy high-risk (e.g., due to hypertension, diabetes) ^ | 21.8% (1,327/6,096) | 23.6% (700/2,963) | 20.0% (627/3,133) |
| # Total pregnancies, including current pregnancy | 2.37 (1.47) | 2.42 (1.51) | 2.32 (1.42) |
| Prior pregnancy high-risk (e.g., complicated by pre-eclampsia, postpartum hemorrhage) ^ | 42.6% (1,722/4,040) | 41.1% (818/1,988) | 44.1% (904/2,052) |
| PHQ-2 score | 1.57 (1.60) | 1.51 (1.58) | 1.63 (1.61) |

^ Indicator variable denoting the % of participants for whom the respective characteristic was present.

Notes: Continuous variables summarized by their sample mean and standard deviation: mean (SD); binary variables summarized by their sample mean as a %, with the respective fraction of participants.

ANC, antenatal care; KES, Kenyan Shilling; PHQ-2, Patient Health Questionnaire-2.

purchasing health insurance. No statistically significant difference in women's plans to breast-feed within 1 h of childbirth was found. Meanwhile, women in the treatment arm were 2.8 ppt (95% CI [−5.4, −0.2]; $p = 0.036$) less likely to arrive at a facility within 2 h of childbirth than women in the control arm.

With respect to routine care seeking, women in the treatment arm had a 0.07 SD (95% CI [0.03, 0.11]; $p = 0.003$) higher summary index compared to those in the control arm (**Table 5**).

**Table 3. Intervention fidelity, ascertained at antenatal follow-up.**

| | Control Mean | Treatment Mean | Unadjusted Treatment Effect [a] | Adjusted Treatment Effect [b] |
|---|---|---|---|---|
| Share of participants who reported receipt of any messages from their county offering pregnancy advice [c] | 0.09 | 0.85 | 0.76 *** 95% CI: (0.73, 0.79) $P < 0.001$ | 0.76 *** 95% CI: (0.73, 0.78) $P < 0.001$ |

* $P < 0.05$

** $P < 0.01$

*** $P < 0.001$.

[a] The unadjusted model only includes covariates that account for the randomization procedure (e.g., recruitment facility's baseline normal vaginal birth volume tertile).

[b] The adjusted model adds baseline individual- and recruitment-facility-level covariates, including maternal age, gestational age, an indicator for first pregnancy, an indicator for secondary school attainment, an indicator for adequate ANC, an indicator for previous receipt of SMS messages from one's county offering pregnancy advice, facility level, and facility county.

[c] Ascertained via phone surveys administered by study enumerators at antenatal follow-up.

ANC, antenatal care; SMS, Short Message Service.

Though no statistically significant difference in the total number of ANC visits was found, the share of women receiving at least the guideline-recommended number of 4 ANC visits was significantly higher (3.1 ppt; 95% CI [0.4, 5.7]; $p = 0.023$) in the treatment arm. As for postpartum care, there was a 4.8 ppt (95% CI [0.7, 9.0]; $p = 0.023$) increase in the share of women who attended at least 1 PNC visit within 6 weeks of childbirth during which their own health was discussed, as well as a 7.4 ppt (95% CI [3.5, 11.3]; $p < 0.001$) increase in the share of women receiving at least the guideline-recommended number of PNC visits. The latter corresponded to an 18% relative increase in the treatment arm. Of note, increases in ANC and PNC utilization appeared to be concentrated around the respective national-guideline-recommended thresholds (**Fig 4**). Meanwhile, no statistically significant differences in the rate of facility-based childbirth or the shares of women seeking medical advice or treatment for antenatal, postpartum, or neonatal concerns were found.

Turning to care seeking in response to danger signs, no statistically significant difference across the treatment and control groups was found in the summary index (**Table 5**). However, the treatment group did have a 2.1 ppt (95% CI [0.4, 3.7]; $p = 0.015$) higher rate of care seeking for postpartum maternal danger signs than the control group.

The intervention led to a 0.09 SD (95% CI [0.07, 0.12]; $p < 0.001$) higher newborn care index (**Table 5**), driven by improvements in newborn sleep positioning and engagement. Specifically, the share of mothers in the treatment group who reported always putting their newborn to sleep at night on their back was 1.9 ppt (95% CI [1.0, 2.8]; $p < 0.001$) higher than in the control group and the share of mothers who reported frequently engaging with their newborn through singing and talking was 5.2 ppt (95% CI [3.2, 7.2]; $p < 0.001$) higher. No statistically significant difference in the rate of exclusive breastfeeding was found, though the control group rate was already 93% (2,462/2,641).

Finally, the intervention led to a 0.06 SD (95% CI [0.01, 0.12]; $p = 0.043$) higher postpartum care content index (**Table 5**). As depicted in **Fig 5**, improvements in the overall domain were driven by impacts on the content of care that mothers—as opposed to their newborns—received during at least 1 postpartum checkup. These chiefly included discussions of mothers' own health, counseling on family planning, and receipt of a physical examination, which had relative increases of 8%, 11%, and 13%, respectively, in the treatment arm compared to the control arm. With respect to newborns, no meaningful differences were found in the frequency of providers conducting newborn physical exams or administering newborn vaccines,

**Table 4. Intervention impact on knowledge and preparedness across the pregnancy-postpartum care continuum.**

| | Control Mean | Treatment Mean | Unadjusted Treatment Effect [a] | Adjusted Treatment Effect [b] |
|---|---|---|---|---|
| **Knowledge** | | | | |
| Summary index [c] | 0.00 | 0.08 | 0.08 ** 95% CI: (0.03, 0.13) $P_{HB}$ = 0.007 | 0.08 ** 95% CI: (0.03, 0.12) $P_{HB}$ = 0.002 |
| # Signs of labor listed without prompting [d] | 1.81 | 2.04 | 0.22 ** 95% CI: (0.08, 0.37) $P$ = 0.002 | 0.24 *** 95% CI: (0.16, 0.32) $P$ < 0.001 |
| Share of antenatal danger sign knowledge questions correctly answered [e] | 0.67 | 0.71 | 0.04 * 95% CI: (0.01, 0.07) $P$ = 0.019 | 0.04 *** 95% CI: (0.02, 0.05) $P$ < 0.001 |
| Share of postpartum danger sign knowledge questions correctly answered [f] | 0.68 | 0.68 | 0.00 95% CI: (−0.01, 0.01) $P$ = 0.584 | 0.00 95% CI: (−0.01, 0.01) $P$ = 0.736 |
| Share of neonatal danger sign knowledge questions correctly answered [g] | 0.64 | 0.65 | 0.00 95% CI: (−0.01, 0.01) $P$ = 0.510 | 0.01 95% CI: (0.00, 0.01) $P$ = 0.146 |
| **Birth Preparedness** | | | | |
| Summary index [h] | −0.02 | 0.07 | 0.09 95% CI: (0.02, 0.16) $P_{HB}$ = 0.057 | 0.08 * 95% CI: (0.02, 0.13) $P_{HB}$ = 0.018 |
| Total # items done in preparation for childbirth [i] | 1.33 | 1.49 | 0.15 95% CI: (−0.01, 0.30) $P$ = 0.059 | 0.13 ** 95% CI: (0.04, 0.21) $P$ = 0.003 |
| Mother had plan to breastfeed within 1 h of childbirth [^] | 0.86 | 0.86 | 0.00 95% CI: (−0.03, 0.03) $P$ = 0.945 | 0.01 95% CI: (−0.02, 0.03) $P$ = 0.666 |
| Late arrival at facility for childbirth [j,^,∀] | 0.25 | 0.22 | −0.04 * 95% CI: (−0.07, 0.00) $P$ = 0.039 | −0.03 * 95% CI: (−0.05, 0.00) $P$ = 0.036 |

* $P$ < 0.05

** $P$ < 0.01

*** $P$ < 0.001, $P_{HB}$: Holm–Bonferroni-adjusted $P$-value.

[^] Indicator variable denoting the share of participants for whom the respective outcome was present.

[∀] An unregistered exploratory outcome.

[a] The unadjusted model only includes covariates that account for the randomization procedure (e.g., recruitment facility's baseline normal vaginal birth volume tertile).

[b] The adjusted model adds baseline individual- and recruitment-facility-level covariates, including maternal age, gestational age, an indicator for first pregnancy, an indicator for secondary school attainment, an indicator for adequate ANC, an indicator for previous receipt of SMS messages from one's county offering pregnancy advice, facility level, and facility county.

[c] Knowledge index composed of 4 preregistered outcomes, namely the shares of antenatal, postpartum, and neonatal danger sign knowledge questions answered correctly and the number of signs of labor listed, without prompting.

[d] See S3 Text for signs of labor assessed.

[e] See S3 Text for antenatal danger signs assessed.

[f] See S3 Text for postpartum danger signs assessed.

[g] See S3 Text for neonatal danger signs.

[h] Birth preparedness index composed of 2 preregistered outcomes, namely the total number of items completed in preparation for childbirth and whether a participant had a plan to breastfeed within 1 h of childbirth and 1 exploratory outcome pertaining to the timeliness of facility arrival for childbirth.

[i] See S3 Text for assessed items to be done in preparation for childbirth.

[j] Late facility arrival defined as arrival within 2 h of childbirth.

ANC, antenatal care; SMS, Short Message Service.

**Table 5. Intervention impact on care seeking, behavior, and content across the pregnancy-postpartum care continuum.**

| | Control Mean | Treatment Mean | Unadjusted Treatment Effect [a] | Adjusted Treatment Effect [b] |
|---|---|---|---|---|
| **Routine Care Seeking** | | | | |
| Summary index [c] | −0.01 | 0.06 | 0.07 95% CI: (0.00, 0.14) $P_{HB}$ = 0.166 | 0.07 ** 95% CI: (0.03, 0.11) $P_{HB}$ = 0.003 |
| Total # ANC visits attended | 4.65 | 4.73 | 0.08 95% CI: (−0.18, 0.34) $P$ = 0.548 | 0.10 95% CI: (−0.04, 0.23) $P$ = 0.161 |
| At least 1 PNC visit attended by mother within 6 weeks of childbirth during which own health was discussed [^] | 0.64 | 0.69 | 0.05 95% CI: (−0.02, 0.12) $P$ = 0.198 | 0.05 * 95% CI: (0.01, 0.09) $P$ = 0.023 |
| Received at least the national-guideline-recommended # ANC visits [d,^,∀] | 0.78 | 0.82 | 0.04 95% CI: (−0.01, 0.08) $P$ = 0.150 | 0.03 * 95% CI: (0.00, 0.06) $P$ = 0.023 |
| Received at least the national-guideline-recommended # PNC visits [e,^,∀] | 0.41 | 0.48 | 0.07 95% CI: (−0.01, 0.14) $P$ = 0.075 | 0.07 *** 95% CI: (0.04, 0.11) $P$ < 0.001 |
| Childbirth occurred in hospital or other formal health center [^] | 0.98 | 0.99 | 0.01 95% CI: (−0.01, 0.03) $P$ = 0.326 | 0.01 95% CI: (0.00, 0.02) $P$ = 0.104 |
| Medical advice or treatment sought during prior month of pregnancy [^] | 0.29 | 0.30 | 0.01 95% CI: (−0.02, 0.05) $P$ = 0.503 | 0.02 95% CI: (−0.01, 0.04) $P$ = 0.248 |
| Medical advice or treatment sought for mother's health postpartum [^] | 0.19 | 0.20 | 0.01 95% CI: (−0.01, 0.04) $P$ = 0.304 | 0.01 95% CI: (−0.01, 0.03) $P$ = 0.396 |
| Medical advice or treatment sought for newborn's health postpartum [^] | 0.38 | 0.40 | 0.02 95% CI: (−0.02, 0.06) $P$ = 0.379 | 0.00 95% CI: (−0.03, 0.03) $P$ = 0.864 |
| **Danger Sign Care Seeking** | | | | |
| Summary index [f] | 0.02 | 0.05 | 0.03 95% CI: (−0.02, 0.09) $P_{HB}$ = 0.267 | 0.04 95% CI: (−0.01, 0.08) $P_{HB}$ = 0.096 |
| Medical care sought for mother in response to ≥1 antenatal danger sign [g,^] | 0.54 | 0.56 | 0.02 95% CI: (−0.02, 0.07) $P$ = 0.340 | 0.03 95% CI: (0.00, 0.06) $P$ = 0.079 |
| Medical care sought for mother in response to ≥1 postpartum danger sign [h,^] | 0.14 | 0.16 | 0.02 95% CI: (0.00, 0.04) $P$ = 0.075 | 0.02 * 95% CI: (0.00, 0.04) $P$ = 0.015 |
| Medical care sought for newborn in response to ≥1 neonatal danger sign [I,^] | 0.22 | 0.23 | 0.01 95% CI: (−0.02, 0.03) $P$ = 0.542 | 0.01 95% CI: (−0.01, 0.03) $P$ = 0.361 |
| **Newborn Care** | | | | |
| Summary index [j] | 0.00 | 0.10 | 0.10 *** 95% CI: (0.06, 0.14) $P_{HB}$ < 0.001 | 0.09 *** 95% CI: (0.07, 0.12) $P_{HB}$ < 0.001 |
| Newborn exclusively breastfed by mother [^] | 0.93 | 0.94 | 0.01 95% CI: (−0.01, 0.03) $P$ = 0.258 | 0.01 95% CI: (0.00, 0.02) $P$ = 0.139 |
| Newborn always put to sleep through the night on their back [^] | 0.03 | 0.05 | 0.02 * 95% CI: (0.00, 0.03) $P$ = 0.039 | 0.02 *** 95% CI: (0.01, 0.03) $P$ < 0.001 |
| Newborn sung/talked to by mother many times over past 24 h [^] | 0.82 | 0.88 | 0.06 *** 95% CI: (0.03, 0.09) $P$ < 0.001 | 0.05 *** 95% CI: (0.03, 0.07) $P$ < 0.001 |
| **Postpartum Care Content** | | | | |

*(Continued)*

**Table 5.** (Continued)

| | Control Mean | Treatment Mean | Unadjusted Treatment Effect [a] | Adjusted Treatment Effect [b] |
|---|---|---|---|---|
| Summary index [k] | 0.00 | 0.06 | 0.06<br>95% CI: (−0.01, 0.14)<br>$P_{HB} = 0.229$ | 0.06 *<br>95% CI: (0.01, 0.12)<br>$P_{HB} = 0.043$ |
| Mother's health discussed with a provider during at least 1 PNC visit [^,∀] | 0.63 | 0.68 | 0.05<br>95% CI: (−0.03, 0.12)<br>$P = 0.208$ | 0.05 *<br>95% CI: (0.00, 0.09)<br>$P = 0.028$ |
| Provider conducted physical exam for mother during at least 1 PNC visit [^,∀] | 0.54 | 0.61 | 0.07<br>95% CI: (−0.01, 0.15)<br>$P = 0.096$ | 0.06 **<br>95% CI: (0.01, 0.11)<br>$P = 0.010$ |
| Provider discussed family planning with mother during at least 1 PNC visit [^,∀] | 0.53 | 0.61 | 0.08<br>95% CI: (−0.01, 0.16)<br>$P = 0.069$ | 0.07 *<br>95% CI: (0.01, 0.14)<br>$P = 0.024$ |
| Provider offered mother cervical cancer screening during at least 1 PNC visit [^,∀] | 0.11 | 0.11 | 0.00<br>95% CI: (−0.03, 0.02)<br>$P = 0.897$ | 0.00<br>95% CI: (−0.02, 0.03)<br>$P = 0.723$ |
| Provider conducted physical exam for newborn during at least 1 PNC visit [^,∀] | 0.95 | 0.96 | 0.01<br>95% CI: (0.00, 0.03)<br>$P = 0.080$ | 0.01 *<br>95% CI: (0.00, 0.02)<br>$P = 0.017$ |
| Provider provided immunization for newborn during at least 1 PNC visit [^,∀] | 0.98 | 0.98 | 0.00<br>95% CI: (−0.01, 0.01)<br>$P = 0.717$ | 0.00<br>95% CI: (−0.01, 0.00)<br>$P = 0.475$ |

* $P < 0.05$

** $P < 0.01$

*** $P < 0.001$, $P_{HB}$: Holm–Bonferroni-adjusted $P$-value.

[^] Indicator variable denoting the share of participants for whom the respective outcome was present.

[∀] An unregistered exploratory outcome.

[a] The unadjusted model only includes covariates that account for the randomization procedure (e.g., recruitment facility's baseline normal vaginal birth volume tertile).

[b] The adjusted model adds baseline individual- and recruitment-facility-level covariates, including maternal age, gestational age, an indicator for first pregnancy, an indicator for secondary school attainment, an indicator for adequate ANC, an indicator for previous receipt of SMS messages from one's county offering pregnancy advice, facility level, and facility county.

[c] Routine care seeking index composed of 2 preregistered outcomes, namely the total number of ANC visits attended and whether a participant received PNC with their own health discussed at least once within 6 weeks of childbirth, and 2 exploratory outcomes pertaining to receipt of the guideline-recommended quantity of routine ANC and PNC.

[d] National guidelines in Kenya recommend at least 4 ANC visits across pregnancy.

[e] Following the first postpartum week, national guidelines in Kenya recommend at least 2 additional PNC visits, between weeks 1–2 and weeks 4–6 postpartum.

[f] Danger sign care seeking index composed of 3 preregistered outcomes, corresponding to each of 3 categories of severe danger signs: antenatal, postpartum, and neonatal. For each category, we assessed whether participants reported seeking medical advice or treatment from a health care provider in response to at least a single danger sign.

[g] See S3 Text for antenatal danger signs assessed.

[h] See S3 Text for postpartum danger signs assessed.

[i] See S3 Text for neonatal danger signs assessed.

[j] Newborn care index composed of 3 preregistered outcomes, namely indicators denoting whether participants reported exclusively breastfeeding, always putting their newborn to sleep through the night on their back, and singing or talking to their newborn many times during the preceding day.

[k] Postpartum care content index composed of 6 exploratory outcomes, namely indicators denoting whether, during at least once PNC visit, participants reported that their health was discussed with a provider, that their provider conducted a physical exam for them, that their provider discussed family planning with them, that their provider offered them cervical cancer screening, that their provider conducted a physical exam for their newborn, and that their provider offered immunizations to their newborn.

ANC, antenatal care; PNC, postnatal care; SMS, Short Message Service.

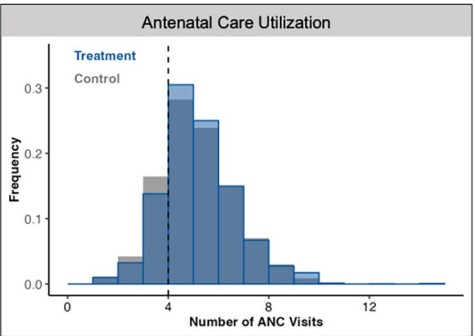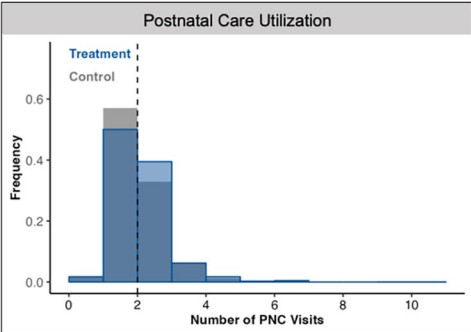

**Fig 4. Antenatal and postnatal care volume, by intervention arm.** This figure displays the increase in volume of ANC visits and PNC visits among participants recruited from treatment facilities just at or above the national-guideline-based thresholds of ≥4 ANC visits throughout pregnancy and ≥2 PNC visits during the first 6 weeks postpartum, following the first postpartum week. ANC, antenatal care; PNC, postnatal care.

with rates of these practices already high in the control group (95% [2,517/2,663] and 98% [2,606/2,663], respectively). No difference across groups was found in the rate of mothers being offered cervical cancer screening either.

We provide additional estimates for preregistered outcomes that are not reported on in the main manuscript in **S6 Table**.

## Sensitivity analysis

As evidenced by **S2 Table**, most of our findings were robust to Holm–Bonferroni *p*-value adjustment, which accounts for the multiple hypotheses tested within each domain. However, several results did not maintain significance at the 5% level after adjusting for multiple

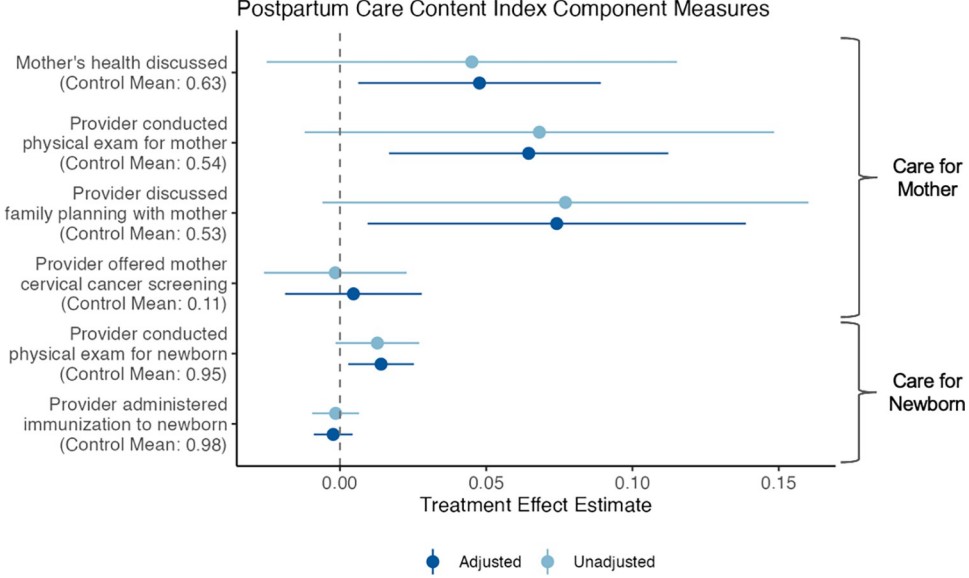

**Fig 5. Impacts on the content of postpartum care received by mothers and their newborns.** This figure depicts adjusted and unadjusted treatment effects for the 6 component measures (all denoting the share of participants for whom the respective outcome was present) incorporated in the postpartum care content index. The figure reveals that overall improvements in the domain were driven primarily by improvements in care that mothers received (e.g., more frequent discussions about their own health and discussions about family planning).

comparisons. These included the estimated reduction in the rate of late facility arrival (i.e., within 2 h of childbirth) and improvements in the care that mothers experienced during post-partum visits (e.g., counseling on family planning, physical examination), which were significant at the 10% level post-adjustment. Meanwhile, the improvements identified in the share of women receiving the guideline-recommended number of ANC visits and the share of women attending at least 1 PNC visit within 6 weeks of childbirth during which their own health was discussed were no longer significant even at the 10% level.

## Discussion

We found that the PROMPTS digital health platform led to a range of improvements across the pregnancy-postpartum care continuum, including in participants' knowledge, birth preparedness, care seeking, newborn care, and postpartum care content. Although the standardized effect sizes that we identified for each index are, in isolation, modest, we found directionally consistent improvements across all domains. Moreover, given the real-world scale of the intervention, even such modest changes can be clinically significant.

Our results revealed particular gains in the postpartum setting, with an emphasis on both mothers and their newborns. Rates of postpartum care increased, both for routine checkups and maternal danger signs, with a nearly 20% rise in the share of women receiving at least the national-guideline-recommended quantity of PNC visits. Moreover, the content of postpartum checkups featured a distinct focus on mothers, with an over 10% increase in rates of family planning counseling and receipt of a physical examination. Of note, maternal care content measures were substantially lower than neonatal measures in the control group, offering an increased opportunity for PROMPTS to influence care for mothers.

While other effects pertaining to knowledge, birth preparedness, and danger sign care seeking were more modest, the increases may still be meaningful. For one, Jacaranda Health estimates that PROMPTS costs 74 cents per participant for their lifetime on the platform (including the costs of message delivery, technical infrastructure, field enrollment, and county engagement), and PROMPTS has achieved notable scale, with over 750,000 individuals enrolled as of November 2021, the start of our study period, and over 2 million additional users since then [36].

Our estimates may also constitute a lower bound of PROMPTS's potential impact. Specifically, our ITT estimates capture the impact of the offer of PROMPTS and do not account for take-up of the platform. Accounting for relative take-up (i.e., scaling our estimates by 1/[treatment group take-up–control group take-up] ≈ 1/0.76 ≈ 31%) may better isolate the impact of adopting the platform. On the other hand, imperfect adoption is to be expected with any intervention, and perhaps the 76 ppt higher platform take-up that we identified is a reasonable indication of real-world adoption. In fact, routinely collected administrative data from Jacaranda Health suggest that only 3/4 of enrollees actively engage with PROMPTS (e.g., by responding to questions administered on the platform or submitting questions of their own).

Considering PROMPTS's real-world scale, it is worth highlighting that improvements in danger sign knowledge and care seeking were quite low, despite these being stated priorities of the intervention. Moreover, while we found evidence of increased rates of guideline-recommended ANC and PNC, we did not identify any changes in the shares of individuals seeking medical advice or treatment for general antenatal, postpartum, or neonatal concerns. Both findings suggest areas for intervention improvement, amidst ongoing efforts to expand PROMPTS.

With respect to our estimated effects, the improvements in knowledge that we identified were concentrated in the antenatal as opposed to the postpartum period, with a 5% increase in

antenatal danger sign knowledge scores and a 13% increase in the number of labor signs listed. This could be attributed to participants' increased attention to PROMPTS message content in the initial phases of enrollment and decreased attention amidst the heightened demands of the postpartum period. Moreover, the ostensible fact that improvements in antenatal knowledge did not translate to as substantial of gains in care seeking could be explained by already high levels of ANC in the control group. Consistent with Kenya's historic emphasis on maternal care during this period, nearly 80% (2,095/2,674) of women in the control group attended at least 4 ANC visits, with over 50% (908/1,690) seeking care for at least 1 antenatal danger sign [8,10,11]. Nevertheless, our findings are consistent with previous studies, which have highlighted the role of mobile health in pregnant and postpartum women's education, including a study in India that found that SMS messages led to improvements in knowledge, particularly for mothers in the antenatal setting [30,45]. Moreover, though PROMPTS may overcome a number of hurdles that health system-level approaches to improving knowledge have faced (e.g., loss of in-person follow-up postpartum, inadequate CHW coverage), neither have been as successful at knowledge improvement in the postpartum period [46–48].

Prior qualitative work has also suggested a role for SMS messages in promoting increased engagement, healthy behaviors, and timely care seeking during pregnancy [49]. Relatedly, we observed a nearly 10% increase in the number of items that women in the treatment arm completed in preparation for childbirth. To the extent that such improvements may have influenced timeliness of care seeking for childbirth, we also observed suggestive evidence of a decrease in the rate of pregnant women arriving at their facility within 2 h of childbirth. Our results on preparedness behaviors align with literature on a specific class of interventions geared towards improving birth preparedness and complication readiness (BPCR). Such interventions are premised on the notion that improvements in birth preparedness (e.g., saving money, arranging transportation) will facilitate timely use of care. While successful, BPCR interventions have historically operated at the health system level, through in-person counseling, or via community mobilization efforts [50].

With respect to routine care seeking, our results are situated in an extensive literature demonstrating the effectiveness of mobile health tools and other interventions targeted towards pregnant and postpartum women, particularly for increasing ANC attendance and facility-based childbirth [30–32]. While prior studies across sub-Saharan Africa have identified larger effect sizes than what we estimate (10 ppt increase in ANC attendance, 7 ppt increase in facility-based childbirth, 20 ppt increase in PNC attendance), those studies were either smaller in scale, conducted in a single site, focused on broader interventions, and/or from settings with lower baseline rates of formal care seeking [51–54]. Moreover, though prior work on the cost-effectiveness of such strategies is limited, our findings align in magnitude with prior health system-level initiatives that have been deemed cost-effective in increasing rates of guideline-recommended ANC [28,55]. PROMPTS also impacts a broader array of domains in a more targeted fashion than initiatives with a lower cost profile (e.g., national media campaigns) [56].

As for danger signs, prior work has suggested difficulty in improving care seeking for maternal illnesses in the absence of comprehensive health system-level support, including CHW home visits, counseling, and danger sign recognition [27]. We found that PROMPTS led to modest increases in care seeking for maternal danger signs, in line with findings from an evaluation of Jacaranda Health's precursor platform to PROMPTS [37]. Nevertheless, we found no such effect on care seeking for neonatal danger signs, which prior health system-level approaches have demonstrated modest success in improving.

Our results on postpartum care content and newborn care are also situated in an emerging literature focused on the postpartum setting. Prior work in sub-Saharan Africa on facility-based provider trainings found similar increases in the rate of family planning counseling as

our study [46]. While encouraging, other work highlights limitations in the extent of content covered through such health system-level strategies, particularly in overburdened delivery settings, and acknowledges the potential of post-discharge SMS outreach [46,48]. In fact, a prior evaluation of Jacaranda Health's precursor platform to PROMPTS found that women who received text messages about family planning were more likely to use postpartum family planning services [37]. Previous evaluations of mobile health interventions have also demonstrated sizable improvements in rates of exclusive breastfeeding [32]. While we identified no such effect, rates of exclusive breastfeeding were high in our control sample (93% [2,462/2,641]), unlike in the other settings evaluated. Indeed, for behaviors with lower baseline adoption in our study (e.g., newborn sleep positioning and engagement), we found notable improvements. While the cost-effectiveness of interventions to improve postpartum care in lower-income settings remains a gap in the literature, our findings suggest a broad role for low-cost mobile health tools like PROMPTS to advance both postpartum and newborn care [28].

## Strengths and limitations

The key strengths of our study were its size and breadth: we recruited and followed over 6,000 individuals across 40 health facilities in 8 Kenyan counties and rather than focus on a narrow set of metrics, we examined several domains across the pregnancy-postpartum care continuum. Moreover, we evaluated a cluster RCT to obtain causal impact estimates.

Nonetheless, our study also had limitations. First, we examined a substantial number of outcomes, which increased the risk of false positive conclusions. However, we mitigated this risk by organizing our outcomes into domains, generating summary indices, and controlling the FWER across indices. Still, our sensitivity analyses suggest that select results for individual index components should be interpreted with caution. Second, all outcomes were self-reported, raising the risk of recall bias or social-desirability bias. Third, aside from capturing the timing of facility arrival before childbirth, we were limited in our ability to assess timeliness of care. In the absence of data on the timing of ANC and PNC visits, we focused on the quantity of visits overall. Fourth, the interaction of PROMPTS with Jacaranda Health's complementary mentorship program could have plausibly influenced outcomes in the childbirth and postpartum settings. Specifically, provider-facing trainings could have influenced women's childbirth experiences and the extent to which they felt empowered to return for postpartum care. However, it is important to note that provider trainings emphasized care for relatively rare emergencies, potentially alleviating this concern. Fifth, the COVID-19 pandemic may have dampened the impact of PROMPTS on care seeking behavior. Nonetheless, by March 2021, over half a year before baseline data collection, 96% of households in Kenya were able to access health services as before the pandemic, and by May 2021, major lockdowns had been lifted [57]. Finally, our study was not sufficiently powered to detect effects on major health outcomes, such as maternal and perinatal mortality.

## Implications and future directions

To our knowledge, our paper is the first to demonstrate the impact of an at-scale digital health platform across the pregnancy-postpartum care continuum in sub-Saharan Africa. While we identified several effects across the continuum, we found notable advances in the postpartum setting, for both mothers and newborns. In almost all cases, the estimated effects were for outcomes based on PROMPTS content sent to enrollees (**Table 1**), with null effects for those without corresponding messages (e.g., postpartum cervical cancer screening). The results have important implications for efforts to improve postpartum care, considering historical under-emphasis on the postpartum setting and the failure of existing systems to adequately deliver

important aspects of care. PROMPTS was developed with this context in mind. It is not a standalone tool or substitute for formal health care; rather, it is intended to be a complement to the established health care infrastructure of Kenya, helping to overcome some of the limitations of often short and hurried clinical visits.

The impacts identified are also notable given the platform's low cost. Considering the lifetime cost per enrollee estimated by Jacaranda Health, our results suggest that 74 cents per individual can lead to a 10% relative increase (0.13/1.33; **Table 4**) in the number of preparatory items done in advance of childbirth and a 17% relative increase (0.07/0.41; **Table 5**) in the share of individuals receiving the guideline-recommended quantity of PNC [36]. Nevertheless, knowledge of and care seeking for danger signs only improved minimally, highlighting an important place for intervention improvement.

Further research is warranted in several areas. Given plans to scale up PROMPTS in settings outside of Kenya, it will be valuable to see how our findings generalize to contexts with different patterns of mobile phone penetration, health literacy, and MNH care seeking behavior. Future research would also be useful to identify subgroups that maximally benefit from the platform and to disentangle the extent to which the impacts we identified are driven by PROMPTS's informational messages versus appointment reminders versus AI-enabled clinical helpdesk. Ongoing research is also examining the potential for synergies of platforms like PROMPTS with health system-level strategies to improve MNH outcomes.

## Supporting information

**S1 Protocol. Initial trial protocol approved by Harvard Longwood Campus Institutional Review Board.**
(PDF)

**S1 Checklist. Consolidated Standards of Reporting Trials (CONSORT) checklist of information to include when reporting a randomized trial, with extensions for cluster-randomized trials.**
(PDF)

**S2 Checklist. CONSORT checklist for abstracts to reports of randomized trials, with extensions for cluster-randomized trials.**
(PDF)

**S1 Text. Technical overview of facility-level randomization.**
(PDF)

**S2 Text. Preregistered outcomes excluded from the main manuscript.**
(PDF)

**S3 Text. Detailed description of knowledge, birth preparedness, and danger sign care seeking outcome measures.**
(PDF)

**S1 Table. Preregistered vs. exploratory outcomes within each outcome domain.**
(PDF)

**S2 Table. Comparison of unadjusted and Holm–Bonferroni-adjusted *P*-values for index component measures.**
(PDF)

**S3 Table. Attrition in baseline study sample during antenatal and postpartum follow-up.**
(PDF)

**S4 Table. Baseline characteristics of eligible and consented cohort that completed antenatal follow-up.**
(PDF)

**S5 Table. Baseline characteristics of eligible and consented cohort that completed postpartum follow-up.**
(PDF)

**S6 Table. Intervention impact on additional preregistered outcomes.**
(PDF)

## Acknowledgments

We are indebted to all the participants who took part in our study and generously shared their time with us. We are also grateful to our study enumerators for meticulously collecting the data on which our evaluation is based. Finally, we thank the full team at Jacaranda Health for their contributions to this study, particularly Jay Patel and Henry Njogu for carefully curating and sharing administrative data to guide our analyses, and Valerie Scott and Lujia Zhang (Harvard T.H. Chan School of Public Health) for their assistance in preparing the study data.

## Author Contributions

**Conceptualization:** Rajet Vatsa, Wei Chang, Sharon Akinyi, Sarah Little, Cynthia Kahumbura, Anneka Wickramanayake, Sathyanath Rajasekharan, Jessica Cohen, Margaret McConnell.

**Data curation:** Rajet Vatsa, Wei Chang, Catherine Gakii, John Mungai.

**Formal analysis:** Rajet Vatsa, Wei Chang.

**Funding acquisition:** Anneka Wickramanayake, Sathyanath Rajasekharan, Jessica Cohen, Margaret McConnell.

**Investigation:** Rajet Vatsa, Wei Chang, Sharon Akinyi, Sarah Little, Catherine Gakii, John Mungai, Cynthia Kahumbura, Anneka Wickramanayake, Sathyanath Rajasekharan, Jessica Cohen, Margaret McConnell.

**Methodology:** Rajet Vatsa.

**Project administration:** Anneka Wickramanayake, Sathyanath Rajasekharan, Jessica Cohen, Margaret McConnell.

**Supervision:** Catherine Gakii, John Mungai, Anneka Wickramanayake, Sathyanath Rajasekharan, Jessica Cohen, Margaret McConnell.

**Visualization:** Rajet Vatsa, Wei Chang.

**Writing – original draft:** Rajet Vatsa.

**Writing – review & editing:** Rajet Vatsa, Wei Chang, Sharon Akinyi, Sarah Little, Catherine Gakii, John Mungai, Cynthia Kahumbura, Anneka Wickramanayake, Sathyanath Rajasekharan, Jessica Cohen, Margaret McConnell.

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
