## [Editor Report · Decision Letter 0]

26 Jun 2024

Dear Dr Vatsa, 

Thank you for submitting your manuscript entitled "Empowering patients across the maternal-newborn care continuum: A cluster randomized controlled trial testing a digital health platform in Kenya" for consideration by PLOS Medicine.

Your manuscript has now been evaluated by the PLOS Medicine editorial staff and I am writing to let you know that we would like to send your submission out for external peer review.

Please re-submit your manuscript within two working days, i.e. by Jun 28 2024 11:59PM. However, please do let us know if you need more time.

Kind regards,

Syba

Syba Sunny, MBBS, MRes, FRCPath

Associate Editor

PLOS Medicine

ssunny@plos.org

---

## [Decision Letter · Decision Letter 1]

26 Sep 2024

Dear Dr Vatsa,

Many thanks for submitting your manuscript "Empowering patients across the maternal-newborn care continuum: A cluster randomized controlled trial testing a digital health platform in Kenya" (PMEDICINE-D-24-01960R1) to PLOS Medicine. The paper has been reviewed by subject experts and a statistician; their comments are included below and can also be accessed here: [LINK]

As you will see, the reviewers highlighted some aspects that will need addressing. After discussing the paper with the editorial team and an academic editor with relevant expertise, I'm pleased to invite you to revise the paper in response to the reviewers' comments. We plan to send the revised paper to some or all of the original reviewers, and we cannot provide any guarantees at this stage regarding publication.

We ask that you submit your revision by Oct 17 2024 11:59PM. However, if this deadline is not feasible, please contact me by email, and we can discuss a suitable alternative.

Don't hesitate to contact me directly with any questions (ssunny@plos.org). 

Best regards, 

Syba 

Syba Sunny, MBBS, MRes, FRCPath 

Associate Editor

PLOS Medicine

ssunny@plos.org

Comments from the academic editor:

The academic editor was supportive of your manuscript. She looked at the reviewers' comments and believed that these could all be addressed. She suggested that your manuscript move forward to the next stage in the process.

Comments from the reviewers: 

Reviewer #1: PMEDICINE-D-24-01960R1

This manuscript reports the results of a cluster randomized trial a digital health platform for maternal-newborn care support in Kenya. It is great to see a randomised evaluation of a new health service which would otherwise be deployed without a robust assessment of its potential impact. Overall, I found the manuscript well written but I have some concerns about the multiplicity of outcomes and how they relate to the sample size calculation.

Major / general comments:

* There seems to be a lack of connection between the power calculation and the choice of outcomes. The power calculation (lines 335-343) mentions a 6.5 percentage point increase in rate of postnatal care receipt whereas the main outcomes are the various indices. It is also not entirely clear how the target of 4,800 women was arrived at. According to my calculation, with 40 clusters, an ICC of 0.01, a base proportion of 61% and an absolute increase of 6.5 percentage point, one needs an average cluster size of 74 participants i.e. a total sample size of 2,960 (with 80% power and 2-sided alpha of 5%). Please clarify/correct. 

* It is not clear what the primary outcome is and, if multiple outcomes were considered at the same (primary) level, how multiplicity was accounted for. Given the number of outcomes at the same level, it would make sense to apply a method to control for multiplicity. I would suggest Holm-Bonferroni or Holm-Sidak correction. This could potentially be applied first across all summary indices (one family), then across sub-indices within each category (e.g. knowledge category = 4 sub-indices)

* Please clarify whether a pre-specified statistical analysis plan was developed/published before unblinding. If so, I would strongly recommend its inclusion as supplemental material.

Minor / specific comments:

* Abstract: I would suggest reporting the p-value for the main composite indices in addition to 95% CI to help assess the strength of evidence

* Please clarify the process, if any, used to ensure that facilities from different study arms were at least 10 kilometres apart. I understand that randomisation was stratified by tertile of monthly volume; however, there is no mention of stratification by geographical location or that randomisation was constrained by distance between facilities. Please also consider adding details of who was responsible for randomisation including allocation concealment procedures (at what point were facilities informed of their random allocation?).

* For recruitment of pregnant women, please clarify whether all eligible women were approached or if some selection process was applied. According to Figure 2, 3,662 and 3,622 women were assessed for participation in the treatment and control facilities respectively. Please provide details around the selection of women "assessed for participation". For example, were all pregnant women who came to the facility assessed? 

* I would suggest adding a table defining all pre-defined and exploratory indices to supplement the text included on lines 278 - 313 of the manuscript. 

* Lines 331-334 indicates that standard errors were clustered by enrolment facility. Was this done by using Generalised Estimating Equations? Please confirm / clarify in the manuscript.

* Please add the Treatment mean to Table 4 and Table 5.

* I find that Figure 4 lacks clarity. First, it is not clear to me why these 3 metrics were picked out of all possible indices. Second, the spacing between bars does not seem to be consistent. It is particularly visible on the top left panel where the number and alignment of bars between 4 and 8 seems odd. Third, I am not convinced that the current display provides enough evidence of an increase in timing between arrival and childbirth as stated in the footnote. Please consider a different display and/or a statistical test to assess this hypothesis. 

* Please consider present all pre-specified indices in a single table. One option could be to combine table 4 and 5 and/or having a separate table for exploratory outcomes i.e. by combining 4.

-Laurent Billot

Reviewer #2: I congratulate the authors for a well written manuscript

it is definitely a daunting task to collect huge data on the participants!

What was the participants response/feedback on receiving frequent messages [10-40 text messages as mentioned in the manuscript]?

what about those pregnant woman who could not read/ were illiterate? were these only text message or audio messages were sent too?

Duration and dates of study should be included

Reviewer #3: This is a well-written article about the effect of a digital health platform on maternal and newborn care outcomes in Kenya. The authors conducted RCT among over 6000 pregnant individuals in 40 health facilities and showed that the digital health interventions improved knowledge, preparedness, routine care seeking, newborn care, and women's postpartum care. Although they did not examine the mortality outcomes, this article shows the importance of using the digital health platform in resource-limited settings. To improve the quality of this article, the authors may revise their articles based on the following comments.

1. L72: "Low- and lower-middle income settings" vs. "low- and middle-income settings in L87 and L128": Are they the same or different?

2. L82: In this line, please specify what it means by "the recommended number of ANC visits. Ref. 39 shows the WHO's recommendation, which recommended the "ANC models with a minimum of eight contacts." Has Kenya followed this model? This article does not seem to follow the WHO's recommendation.

3. L84 and other places: Please check the reference numbers; some were not written with superscript.

4. Regarding the use of the term "patients," I have many comments.

L112, L123: In these two paragraphs, "digital health tools" are much more emphasized as the tools for "patients" rather than for "healthy pregnant women and mothers," though they were also mentioned a little. It seems better to tone down this "patient" description.

L139: What does "a patient-level solution" mean? Was the target population of this article "patients"?

L146 and other places: All the pregnant women seem to be defined as "patients" in this article, even if some had no danger signs or symptoms such as anemia.

L158~: "Patients" were used in treatment facilities. "Participants" and "women" were used in control facilities. It suggests the target population in the treatment facilities had some medical problems, and those in the control facilities had no such issues. In this article, PROMPS stands for "Promoting Mothers in Pregnancy and Postpartum Through SMS." Must all the mothers who can join PROMPS "patients"?

L201: Eligibility: Must pregnant women be patients in the treatment group?

L240, L244: Were they patients?

L334: Not all the participants were "patients" from this description. Please reconsider the use of "patients" throughout the manuscript, including the introduction contents.

5. L132: MNH nonprofit organization?

6. L136-138: What was the coverage of health facilities and individuals (pregnant women and mothers?) when this study started? Were both coverages increased without having the evidence shown in this article? If so, what is the value of this article?

7. L140: Again, please describe "the guideline-recommended." Which guideline by who?

8. L144: For international readers, please briefly describe Kenya's health system and related MNH policies.

9. L156: How did you know "the other interventions" influenced or not influenced this evaluation?

10. L170: In this paragraph, do participants mean patients?

11. L211: 100 Kenyan shillings. How much is it in US dollars?

12. L229: Please clarify the primary and secondary outcomes in the abstract and the main text. In this place, the primary and secondary needs to be explained.

13. L253: Total number of ANC and PNC visits: What is the policy about them? Again, in Ref 39 (WHO), WHO recommended eight ANC visits. 

14. L258: Considering stagnant neonatal mortality rates in Kenya, the first PNC (usually within 24 or 48 hours after birth) might be the most critical PNC visit. However, in this article, 1st and 2nd PNC visits and 2nd and 3rd PNC visits are considered the same. Is there data to show 1st and 2nd or 1st and 3rd, for example?

15. L270: Were the weight of the 1st, 2nd, and 3rd PNC visits the same?

16. L287: Each ANC visit has the best timing for it. "Adequate prenatal care" means only the number of ANC visits, and its timing was not counted in this article.　

17. L333: If participants do not have their own mobile phones, how did they have access to mobile phones? Were there any differences between the own-phone users and non-users?

18. L339, 340: Participants are equal to respondents?

19. L346: Over 85 % received the messages in the intervention group, and about 10 % received them in the control group. Were there any differences in outcomes between those who received and those who did not receive in each group? Were there any similarities in outcomes between those who received them in both groups?

20. L403, L415: These paragraphs must be the highlight of this article. They can be strengthened by using more references.

21. L412, L512: "Mere 74 cents" seems reasonable to justify the feasibility. Considering sustainability by the Kenyan government or society, is it unnecessary to consider consultancy, management, and other implementation costs for this platform?

22. L413,414: The PROMPTS has been implemented at a remarkable scale in a few years. Based on which evidence was it scaled? Supposing it was scaled without the evidence of this article, what would be the value of this article? 

23. L441: Is it relevant to use Ref 46 in this paragraph if the target population of this article was not patient in general?

24. L490: In L156, it was written that "the other interventions" did not influence the outcome of this evaluation. "Jacaranda Health's complementary mentorship program" was not one of "the other interventions"?

25. L513: "Platform's low cost" sounds like the total cost considering total management and implementation. It must be part of the cost of its implementation. 

26. L562: Inside the figure, "Treatment" was written in black letters, and "control" was written in green letters. The title of Fig. 4 was about the Treatment Arm. What does it mean by "treatment" and "control" in this figure?

27. L726: References: Some reference names were fully spelled out, and some reference names were abbreviated. Please make it consistent.

---

* Please upload any figures associated with your paper as individual TIF or EPS files with 300dpi resolution at resubmission; please read our figure guidelines for more information on our requirements: http://journals.plos.org/plosmedicine/s/figures. While revising your submission, please upload your figure files to the PACE digital diagnostic tool, https://pacev2.apexcovantage.com/. PACE helps ensure that figures meet PLOS requirements. To use PACE, you must first register as a user. Then, login and navigate to the UPLOAD tab, where you will find detailed instructions on how to use the tool. If you encounter any issues or have any questions when using PACE, please email us at PLOSMedicine@plos.org.

* Thank you for providing a Data Available Statement. PLOS has a strong stance that requires data to be available where at all possible. Could the authors possibly be able to provide de-identified or anonymised data? Further information that might be helpful in constructing your response can be found here: https://journals.plos.org/plosmedicine/s/data-availability#loc-human-research-participant-data-and-other-sensitive-data

* Thank you for providing a completed CONSORT checklist. Please add the following statement, or similar, to the Methods: "This study is reported as per [XXXX] guideline (S1 Checklist)."

FIGURES AND TABLES

SUPPLEMENTARY MATERIAL

REFERENCES

RCTs 

* Please structure the Methods section using the following sub-headings: Study design and participants, Randomization and masking, Procedures, Outcomes, Statistical analysis.

* Please ensure that all prespecified outcomes (primary, secondary, and exploratory) are listed in the Methods/Outcomes section and indicate whether there are outcomes that are not presented in the current report.

* Please specify the dates (Month Day, Year) during which study enrollment and follow up occurred.

* Please include absolute numbers wherever you report percentages; eg, n/N (%)

* Please present the safety data for the study including numbers of specific events and whether or not adverse events are thought to be related to treatment. AEs should be reported in the abstract, per CONSORT and CONSORT-Harms.

* If your trial had to undergo important modifications in response to extenuating circumstances, please complete the CONSERVE-CONSORT checklist and provide in your Supporting Information; (https://www.equator-network.org/reporting-guidelines/guidelines-for-reporting-trial-protocols-and-completed-trials-modified-due-to-the-covid-19-pandemic-and-other-extenuating-circumstances-the-conserve-2021-statement/). When completing the checklist, please use section and paragraph numbers, rather than page numbers.

* In keeping with our commitment to Open Science, please include the study protocol document and analysis plan (including any amendments) as Supporting Information to be published with the manuscript if accepted.

* Please note that PLOS Medicine requires prospective, public registration of a data sharing plan (as part of mandatory clinical trials registration) for all clinical trials that began enrollment on or after January 1, 2019, in accordance with ICMJE requirements.

---

## [Decision Letter · Decision Letter 2]

16 Dec 2024

Dear Dr. Vatsa,

Thank you very much for re-submitting your manuscript "Empowering women across the pregnancy-postpartum care continuum: A cluster randomized controlled trial testing a digital health platform in Kenya" (PMEDICINE-D-24-01960R2) for review by PLOS Medicine.

I have discussed the paper with my colleagues and the academic editor and it was also seen again by two reviewers. I am pleased to say that provided the remaining editorial and production issues are dealt with we are planning to accept the paper for publication in the journal.

[LINK]

We look forward to receiving the revised manuscript by Jan 03 2025 11:59PM.   

Sincerely,

Rebecca Kirk

On behalf of:

Syba Sunny, MBBS, MRes, FRCPath

Senior Editor 

PLOS Medicine

plosmedicine.org

Requests from Editors:

GENERAL EDITORIAL REQEUSTS

* Please confirm that your title complies with to PLOS Medicine's style. Your title must be nondeclarative and not a question. It should begin with main concept if possible. "Effect of" should be used only if causality can be inferred, i.e., for an RCT. Please place the study design ("A randomized controlled trial," "A retrospective study," "A modelling study," etc.) in the subtitle (ie, after a colon).

* Please confirm that your abstract complies with our requirements, including providing all the information 

relevant to this study type https://journals.plos.org/plosmedicine/s/submission-guidelines#loc-abstract

* Please ensure that the Introduction ends with a clear description of the study question or hypothesis.

* Please ensure that all abbreviations are defined at first use throughout the text.

FUNDING STATEMENT

* The funding statement should include: specific grant numbers, initials of authors who received each award, URLs to sponsors’ websites. Also, please state whether any sponsors or funders (other than the named authors) played any role in study design, data collection and analysis, the decision to publish, or preparation of the manuscript. If they had no role in the research, include this sentence: “The funders had no role in study design, data collection and analysis, decision to publish, or preparation of the manuscript.”

* It appears that one or more study authors is affiliated with one or more of the agencies that funded the study. Thus, the statement “The funders had no role in study design, data collection and analysis, decision to publish, or preparation of the manuscript” does not apply. Please revise the Financial Disclosure accordingly, as in "[Author name] is [author's role] at [funding agency]. The funders had no other role in study design…..” 

COMPETING INTERESTS STATEMENT

* All authors must declare their relevant competing interests per the PLOS policy, which can be seen here: https://journals.plos.org/plosmedicine/s/competing-interests For authors with ties to industry, please indicate whether any of the interests has a financial stake in the results of the current study.

FIGURES

* Please confirm that the appropriate usage rights apply to the use of this map. Please see our guidelines for map images: https://journals.plos.org/plosmedicine/s/figures#loc-maps

CLINICAL TRIALS

* The sample size listed in the submitted manuscript and the trial registry differ. Please explain the discrepancy.

* The CONSORT flowchart should be figure 1, please revise.

Comments from Reviewers:

Reviewer #1: All my previous comments have been adequately addressed.

-Laurent Billot

Reviewer #3: Almost all my questions were well addressed. I have only a few minor comments.

1. On L117, antenatal care is abbreviated as ANC. The same was done on L353 and antenatal care is used several times without abbreviation (L156, for example). On L210, prenatal care appears but is not abbreviated as PNC.

2. In Tables 4 and 5, asterisks are used for p-value. Is it necessary? The P-value is written in the same tables.

3. For questions 28,29 and 30, I understood the lack of data about ANC and PNC. Your response sounds reasonable, but I wonder if it should be included as a limitation as the timing of these cares is crucial.

[LINK]

---

## [Editor Report · Decision Letter 3]

9 Jan 2025

Dear Dr Vatsa, 

On behalf of my colleagues and the Academic Editor, Annettee Nakimuli, I am pleased to inform you that we have agreed to publish your manuscript "Impact evaluation of a digital health platform empowering Kenyan women across the pregnancy-postpartum care continuum: A cluster randomized controlled trial" (PMEDICINE-D-24-01960R3) in PLOS Medicine.

PRESS

Sincerely, 

Rebecca Kirk

On behalf of:

Syba Sunny, MBBS, MRes, FRCPath 

Senior Editor 

PLOS Medicine